# Crosstalk within a functional INO80 complex dimer regulates nucleosome sliding

Oliver Willhoft, Elizabeth A McCormack, Ricardo J Aramayo, Rohan Bythell-Douglas, Lorraine Ocloo, Xiaodong Zhang, Dale B Wigley*

Department of Medicine, Section of Structural Biology, Imperial College London, London, United Kingdom

**Abstract** Several chromatin remodellers have the ability to space nucleosomes on DNA. For ISWI remodellers, this involves an interplay between H4 histone tails, the AutoN and NegC motifs of the motor domains that together regulate ATPase activity and sense the length of DNA flanking the nucleosome. By contrast, the INO80 complex also spaces nucleosomes but is not regulated by H4 tails and lacks the AutoN and NegC motifs. Instead nucleosome sliding requires cooperativity between two INO80 complexes that monitor DNA length simultaneously on either side of the nucleosome during sliding. The C-terminal domain of the human Ino80 subunit (Ino80CTD) binds cooperatively to DNA and dimerisation of these domains provides crosstalk between complexes. ATPase activity, rather than being regulated, instead gradually becomes uncoupled as nucleosome sliding reaches an end point and this is controlled by the Ino80CTD. A single active ATPase motor within the dimer is sufficient for sliding.

## Introduction

Chromatin remodelling complexes are important regulators of the chromatin landscape and have vital roles in transcription, DNA damage repair and replication (*Clapier and Cairns, 2009*). Many of these complexes vary in complexity from one or a few subunits (e.g. CHD1 [*Delmas et al., 1993*] or ACF [*Ito et al., 1997*]) to larger systems with more than a dozen subunits (e.g. INO80 [*Shen et al., 2000*] and RSC [*Cairns et al., 1996*]). However, despite this variability, these complexes overlap in their basic biochemical activity in carrying out ATP-dependent nucleosome sliding. Precisely how the 147 base pair wrap slides around the histone core is still largely a mystery and regulation of the process is even more poorly understood. It is also unclear which aspects of mechanism are conserved between these highly variable systems and which ones are specific to different complexes.

Chromatin remodelling complexes have been divided into four families (*Clapier and Cairns, 2009*). Two families are somewhat related and comprise either a single subunit (Chd1 family) or a few subunits (ISWI family). By contrast, the SWI/SNF and INO80 families are larger (>1 MDa), multi-subunit complexes that, in addition to a superfamily 2 helicase-like motor subunit (*Singleton et al., 2007*), contain actin and actin-related proteins (ARPs) (*Szerlong et al., 2008*) and other subunits of unknown function. Some complexes, like RSC, slide nucleosomes off DNA ends (*Cairns et al., 1996*) and use ARPs to regulate coupling (*Clapier et al., 2016*). However, others such as ISWI and INO80 slide nucleosomes away from DNA ends (*Hamiche et al., 1999*; *Längst et al., 1999*; *Shen et al., 2000*) and are also able to space nucleosomes evenly on DNA (*Yang et al., 2006*; *Racki et al., 2009*; *Udugama et al., 2011*). The mechanism for spacing of nucleosomes by ISWI family enzymes has been shown to involve a complex interplay between the H4 tails of the nucleosome (*Clapier et al., 2001*), two conserved regulatory sequences (called AutoN and NegC) (*Clapier and*

*For correspondence: d.wigley@imperial.ac.uk

Competing interests: The authors declare that no competing interests exist.

*Cairns, 2012*) and flanking DNA (*Dang et al., 2006*). These components interact to regulate ATPase activity and sense the length of flanking DNA (*Hwang et al., 2014*). Recent structural studies have provided molecular details of the H4 tail interaction with the motor domains of ISWI and suggest how this regulates ATPase activity (*Yan et al., 2016*). Nucleosome sliding by ACF requires cooperative activity between two complexes (*Racki et al., 2009*) and involves different nucleotide-dependent conformational states of the SNF2 subunit in each complex (*Leonard and Narlikar, 2015*). However, the molecular detail of how all of these components (H4 tails, AutoN, NegC, ATPase sites, differential subunit conformations, flanking DNA) interact to coordinate spacing of nucleosomes is still not fully understood.

The INO80 complex can slide nucleosomes in an ATP-dependent manner (*Shen et al., 2000*), and a minimal core of the human complex (hINO80) comprising nine subunits can catalyse this reaction *in vitro* (*Chen et al., 2011*; *Willhoft et al., 2016*). In addition, INO80 is able to centre mono-nucleosomes on short (<250 bp) DNA fragments in vitro (*Jin et al., 2005*; *Udugama et al., 2011*; *Willhoft et al., 2016*) and to space multiple nucleosomes on longer DNA fragments (*Udugama et al., 2011*). However, the Ino80 subunit lacks AutoN and NegC motifs (*Clapier and Cairns, 2012*) and is not regulated by binding of H4 tails (*Udugama et al., 2011*). Consequently, the mechanism by which the complex is able to determine the centre of a DNA fragment is unknown but cannot be the same as in ISWI.

Structural studies on remodelling complexes have been limited to crystal structures of fragments of ISWI and CHD1 family enzymes (*Grüne et al., 2003*; *Hauk et al., 2010*; *Sharma et al., 2011*; *Yamada et al., 2011*; *Yan et al., 2016*) and some subunits of the larger, multi subunit complexes (particularly ARPs [*Fenn et al., 2011*; *Gerhold et al., 2012*; *Saravanan et al., 2012*; *Schubert et al., 2013*; *Lobsiger et al., 2014*; *Cao et al., 2016*). Low resolution three-dimensional electron microscopy structures of SWI/SNF (*Smith et al., 2003*), RSC (*Chaban et al., 2008*), and INO80 (*Tosi et al., 2013*) all suggest monomers of these complexes bound to nucleosomes, whereas smaller complexes such as ACF (*Racki et al., 2009*) and Chd1 (*Nodelman et al., 2017*) show two complexes bound to a nucleosome. Curiously, ACF (ISWI family) has been shown to slide and space nucleosomes as a cooperative dimer (*Racki et al., 2009*) whereas CHD1 appears to work as a monomer (*McKnight et al., 2011*; *Patel et al., 2013*). The oligomeric state of SWI/SNF and INO80 family complexes is presumed to be monomeric based on structural studies (*Smith et al., 2003*; *Chaban et al., 2008*; *Tosi et al., 2013*).

Here we show that although the hINO80 complex, like ACF, appears to be able to centre nucleosomes on DNA fragments, this ability is in fact due to being able to sense when the nucleosome is within 50 base pairs (bp) from a DNA end. When provided with nucleosomes with sufficiently long overhangs, it simply positions them no closer than 50 bp from each end. This property suggests an ability to sense DNA flanking both sides of the nucleosome simultaneously. We show that, like in ACF (*Racki et al., 2009*), nucleosome sliding activity requires cooperativity between two hINO80 complexes, that a single complex is insufficient to promote sliding, but that ATPase activity is necessary within only one of these complexes for sliding to occur. However, this is where any similarity with ACF ends. Instead of an interplay between H4 tails and the AutoN and NegC motifs that regulates ATPase activity and hence sliding, INO80 lacks these motifs and operates by a distinct mechanism. Although the ATPase activity is highly stimulated by binding of the complex to a nucleosome, the rate is unaltered whether the nucleosome is sliding or has reached the central position showing that sliding is not regulated by ATPase activity per se, as seen in ACF, but instead by the extent of coupling of ATPase activity to sliding. Finally, we show that cooperative interactions between the C-terminal domains of the Ino80 subunits between complexes senses DNA flanking lengths and regulates this coupling of ATPase to sliding in order to space nucleosomes. These findings provide insight into how hINO80 remodels nucleosomes and may have implications for regulation and activity of other large, multi-subunit nucleosome remodellers.

## Results

### hINO80 positions nucleosomes 50 bp from ends

Several chromatin remodelling complexes, including INO80, are able to reposition mono-nucleosomes from the end to the centre of DNA fragments (*Shen et al., 2000*; *Stockdale et al., 2006*;

*Udugama et al., 2011*) or evenly space multiple nucleosomes in arrays (*Längst et al., 1999*; *Udugama et al., 2011*). This can be visualised using gel-based methods, which differentiate between nucleosome species based on their electrophoretic mobility (*Figure 1A*). As observed previously for the full yeast (*Shen et al., 2000*; *Udugama et al., 2011*) and human (*Chen et al., 2011*) INO80 complexes, a core complex of human INO80 complex, that we have previously characterised (*Willhoft et al., 2016*) (*Figure 1—figure supplement 2*), is also able to slide nucleosomes from an end position to one at the centre of DNA fragments when the DNA overhang is 100 bp or less (visualised as a single band on gels [*Figure 1A*]). However, for nucleosomes with flanking DNA greater than 100 bp, the number of bands increases as the length of the flanking DNA is increased

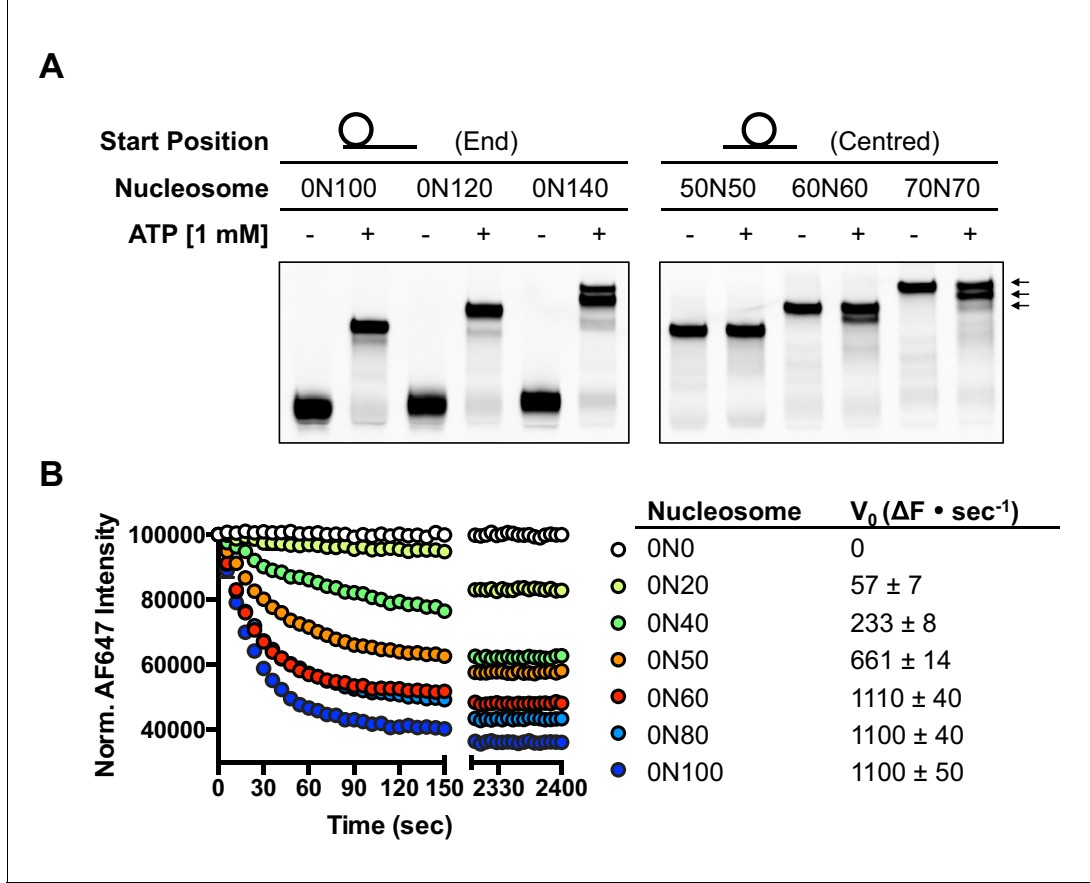

Figure 1. hINO80 positions nucleosomes 50 base pairs from DNA ends. (A) Nucleosomes positioned either at the end (left panel) or centre (right panel) of DNA fragments were incubated with hINO80. As reported previously (*Udugama et al., 2011*) overhangs of 100 bp or less resulted in centrally positioned nucleosomes, while those positioned at the centre remained centrally located. (This property, as well as others (nucleosome binding affinity, sliding activity and ATPase), are also observed for recombinant human INO80 complex containing a full length Ino80 subunit and additional (non-conserved) subunits [*Figure 1—figure supplement 1*]). Increasing the overhang from 100 to 120 bp produced two bands corresponding to a centrally positioned nucleosome and 10 bp from the centre. Increasing to a 140 bp overhang gave three bands – centred, 10 bp from centre and 20 bp from centre (marked with arrows). Incubation of the appropriate centrally positioned nucleosomes gave similar products. (B) (left) FRET-based sliding assay for nucleosomes with different flanking DNA lengths, (right) initial rates of sliding for nucleosomes with different flanking DNA lengths. Comparison with gel-based assay is presented in *Figure 1—figure supplement 2*.

The following figure supplements are available for figure 1:

**Figure supplement 1.** Characterisation of full length recombinant human INO80 complex.

**Figure supplement 2.** Characterisation of hINO80 complex.

**Figure supplement 3.** Nucleosomes shifted by hINO80 are located close to the centre of a short DNA fragment.

(*Figure 1A*). Using the propensity of the Widom 601 sequence (*Lowary and Widom, 1998*) to adopt 'preferred' positions spaced approximately 10 bp apart, we used heat-induced mobilisation of our nucleosome substrate to visualise these preferred positions (*Figure 1—figure supplement 2*). This allowed us to estimate the positional spread of our nucleosome sliding reaction products, with one additional band in the product for every additional 20 bp of flanking DNA at the start of the reaction. Similarly, hINO80 complex cannot slide nucleosomes that are positioned centrally on fragments unless the length of the flanking DNA on both sides exceeds 50 bp. At this point the spread of bands resembles that resulting from repositioning of the appropriate end-positioned nucleosome, showing that both reactions reach the same end point (*Figure 1A*). The simplest explanation of these data is that rather than determining the centre of a DNA fragment, hINO80 is instead able to monitor the length of the DNA flanking the nucleosomes, and this monitoring ability appears to extend to approximately 50 bp on either side. If the flanking DNA is less than 50 bp on either side then a compromise is reached to position the nucleosome approximately equidistant from each end. Although our gel system cannot resolve positions with base pair resolution, the spread of bands is certainly limited to within a few base pairs of this central location (*Figure 1—figure supplement 3*). Consistent with this interpretation, we observe that the sliding rate is proportionately reduced with overhangs shorter than 50 bp (*Figure 1B*). Similar activity has been reported for the ACF remodeller (*Yang et al., 2006*). This reduction in rate explains why hINO80 appears to centre nucleosomes on short (<250 bp) DNA fragments because it gets rapidly and progressively slower as it approaches an end. Since flanking DNA on both sides appear to be monitored simultaneously, we investigated whether a nucleosome could interact with two hINO80 complexes.

## hINO80 binds nucleosomes as a cooperative dimer

We used microscale thermophoresis (MST) to investigate the binding of hINO80 to a variety of different nucleosomes (*Figure 2*). At low nucleosome concentrations, the data revealed that binding was highly cooperative suggesting binding of multiple hINO80 complexes, with a Hill coefficient close to 2 (*Figure 2A*).

We then conducted similar experiments under saturating conditions to enable us to determine stoichiometry of the complexes (*Figure 2B* and *Figure 2—figure supplement 1*). Nucleosomes with either no overhangs, a single 50 bp overhang (0N50) or centrally positioned with 50 bp flanking both sides (50N50), can each bind two hINO80 complexes. We could also detect two separate hINO80 binding events to nucleosomes by gel shift using a centrally-positioned (50N50) nucleosome (*Figure 2—figure supplement 1*). Finally, we visualised hINO80 complexed with nucleosomes directly using negative stain electron microscopy (*Figure 2C*). Although there was significant heterogeneity in particles, most appear to comprise two hINO80 complexes bound to each nucleosome. We conclude that a nucleosome can accommodate two complexes of hINO80 simultaneously and that the large Hill coefficient indicates tight coupling of binding of these complexes. Although the small remodeller ACF has been shown to act as a dimer (*Racki et al., 2009*), the finding that the much larger hINO80 complex, which is considerably larger than nucleosomes themselves, also acts as a dimer was unanticipated. Structural studies to date all suggest large remodelling complexes such as SWI/SNF (*Smith et al., 2003*), RSC (*Chaban et al., 2008*) and even INO80 (Tosi et al., 2013) bind to nucleosomes as a monomer. How two such large INO80 complexes can be accommodated around a single nucleosome remains to be determined. Furthermore, this finding has major implications for assays on enzyme activity in which catalytic (sub-stoichiometric) amounts of INO80 complex are used and also in the preparation of samples for structural studies.

## hINO80 dimers slide nucleosomes cooperatively

We next measured nucleosome sliding and ATPase activity to determine the effect of protein concentration on these activities (*Figure 3* and *Figure 3—figure supplement 1*). We used high nucleosome concentrations to allow us to access conditions to determine stoichiometry. At low, sub-stoichiometric (i.e. catalytic), protein concentrations, a small portion of the nucleosomes are processively remodelled but the reaction eventually gets to completion (*Figure 1—figure supplement 1*). However, rather than increasing linearly with protein concentration, the sliding activity instead increases in a sigmoidal fashion, reaching a plateau at a ratio of two hINO80 complexes per nucleosome (*Figure 3A*). The ATPase activity also peaks at a 2:1 ratio of complex to nucleosomes

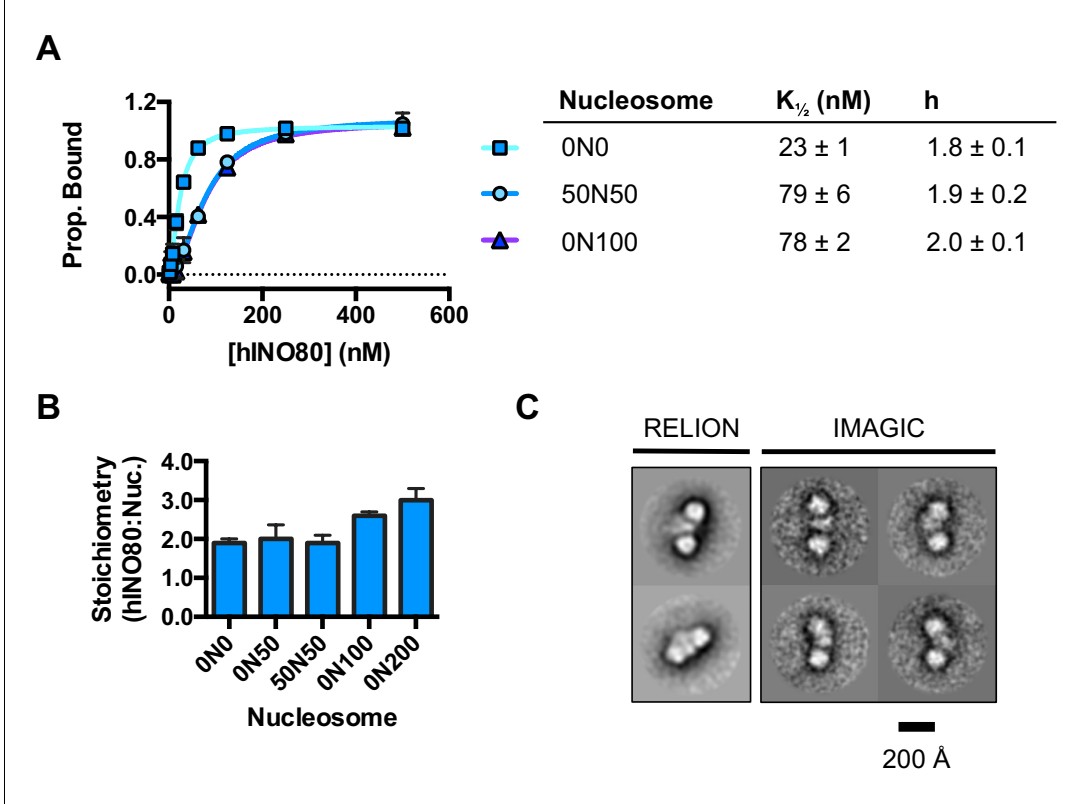

**Figure 2.** hINO80 binds nucleosomes cooperatively. (**A**) Equilibrium binding between hINO80 and nucleosomes determined by MST using 20 nM nucleosome ligands. The data were fitted to a cooperative binding curve (left) to determine $K_{1/2}$ saturation points and Hill coefficients (**h**) (right). Raw data and analysis are presented in *Figure 2—figure supplement 1*. (**B**) Stoichiometry of hINO80 binding to a variety of nucleosomes. In all experiments, the nucleosome concentration used was 500 nM. The 0N100, 0N0 and 50N50 nucleosomes each bind two hINO80 complexes. However, lengthening of the overhang provides an adventitious binding site until, at 200 bp, three hINO80 complexes can be accommodated per nucleosome. Data analysis is presented in *Figure 2—figure supplement 2*. (**C**) Negative stain electron microscopy of hINO80 complexed with 50N50 nucleosomes. The 2D classes show evidence for two hINO80 complexes flanking a nucleosome. RELION classes (shown on the left) were used to select principal views and contained most of the particles, from which smaller groups were selected for classes in IMAGIC (shown on the right). The IMAGIC classes contained fewer particles but which were more closely aligned in view so show more detail.

The following figure supplements are available for figure 2:

**Figure supplement 1.** Raw data and fits for hINO80:nucleosome interactions shown in *Figure 2A*.

**Figure supplement 2.** Characterisation of the hINO80:nucleosome interaction.

(*Figure 3B*). However, although the sliding activity is highly cooperative (h = 1.9 ± 0.1), the ATPase activity is considerably less so (h = 1.3 ± 0.1). By contrast, in the absence of a nucleosome substrate, the ATPase activity shows a linear dependence on protein concentration with no sign of cooperativity across the same concentration range (*Figure 3C*). Taken together, these data demonstrate that a cooperative interaction between two hINO80 complexes is required to both bind to and slide a nucleosome.

Once the sliding reaction reaches an apparent equilibrium end-point, there are three possibilities: (i) ATPase activity ceases and hence sliding is halted, (ii) sliding continues in a limited fashion as the nucleosome oscillates in position around the centre point, or (iii) the complexes become uncoupled from sliding but continue hydrolysing ATP. To resolve these possibilities, we monitored nucleosome sliding and ATPase activities simultaneously using a 2:1 ratio of hINO80:nucleosomes under saturating concentrations (*Figure 3D*). Under these conditions, the ATPase activity continues after sliding has taken the nucleosome beyond the distance detectable in our FRET assay but the same reaction

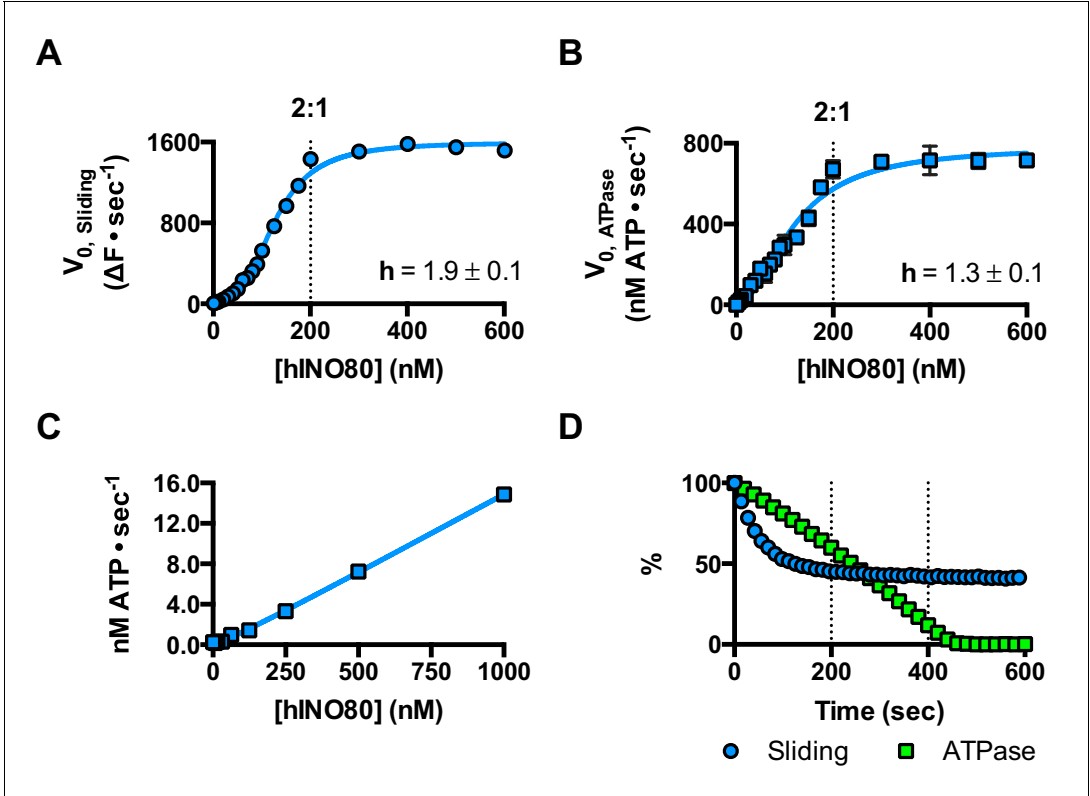

**Figure 3.** hINO80 operates as a functional dimer. (**A**) Sliding activity of nucleosomes increases sigmoidally (h = 1.9 ± 0.1) (*Figure 3—figure supplement 1*) with respect to hINO80 concentration, peaking at a ratio of two hINO80 complexes per 0N100 nucleosome. Experiments were done at a nucleosome concentration of 100 nM. Experiments at 300 nM nucleosome show similar activity profile again peaking at a ratio of 2:1 hINO80: nucleosome (*Figure 3—figure supplement 1*). (**B**) ATPase activity follows a similar profile as sliding activity, reaching a maximum at the same ratio of hINO80:nucleosome, but with a lower cooperativity (h = 1.3 ± 0.1) (*Figure 3—figure supplement 1*). (**C**) ATPase activity of hINO80 shows a linear dependence on protein concentration in the absence of nucleosomes. (**D**) Normalised fluorescence traces corresponding to sliding and ATPase activities of hINO80, at a 2:1 ratio of complex to 0N100 nucleosomes. ATPase activity (determined from the slope) shows a similar rate even after sliding has reached completion (<200 s). Further analysis of these data with 50N50 nucleosomes is presented in *Figure 3—figure supplement 2*.

The following figure supplements are available for figure 3:

**Figure supplement 1.** Hill plots and stoichiometry for sliding and ATPase.

**Figure supplement 2.** ATPase during and after sliding.

monitored by native gel electrophoresis shows that >90% of the end-positioned nucleosome has reached its end point by 90 s (*Figure 3—figure supplement 2*). However, the ATPase activity continues at a similar (or even faster) rate after this point. Furthermore, a centrally positioned nucleosome (50N50), which does not slide from this central position (*Figure 1*), stimulates ATPase to a similar extent as tailed nucleosomes (0N100) (*Figure 3—figure supplement 2*). Similar results were obtained for yeast INO80 with a centrally positioned 53N53 nucleosome that stimulates ATPase but was also shown not to slide, using a method at close to base pair resolution (*Udugama et al., 2011*). By contrast, a centrally positioned 33N33 nucleosome fails to stimulate ATP hydrolysis in ISWIa (*Gangaraju and Bartholomew, 2007*). Consequently, we can exclude the possibility that the ATPase slows or stops once the nucleosome stops sliding for Ino80. This experiment also provides some insight into the differences in the apparent cooperativity seen in the sliding and ATPase kinetics. Although sliding appears to have an absolute requirement for a dimer (with a Hill coefficient of close to 2.0), the ATPase activity comprises a significant component that is uncoupled. In fact, the number of ATP molecules hydrolysed during the first 90 s of the reaction is around eight times more than

the number of base pairs that each nucleosome slides. Furthermore, as the sliding reaction approaches equilibrium, the rate of sliding slows but the ATPase continues unabated. These data show that either productive sliding becomes increasingly uncoupled from ATPase as it approaches its end point, or that the sliding oscillates either side of this midpoint. For longer DNA overhangs, this is indeed the case when the centre point cannot be located precisely (see *Figure 1*). However, for shorter overhangs, we wished to investigate whether the complex oscillates or is stationary and uncoupled. To do this, we designed a series of nucleosome standards that were displaced in 2 bp intervals on one side of the centre point. By running these slowly on a higher percentage gel, we were able to resolve these to determine nucleosome positioning to within 2 bp of the centre (*Figure 1—figure supplement 3*). A reaction left to run to completion shows a very limited spread of bands. Although this does not preclude very rapid oscillations of 1 or 2 bp, we should expect to see some sign of this on our gel. The data therefore suggest that any such oscillation is minimal compared to the ATPase rates so most, possibly all, of the ATPase we observe is uncoupled from sliding.

## A single active ATPase activity is necessary and sufficient for sliding and monitoring

Since nucleosomes appear to come to a slow stop resulting from increased uncoupling of ATPase from sliding, we wished to test the role of the ATPase activity in each monomer during this process. We prepared a mutant hINO80 complex in which the motor domain of the Ino80 subunit carried a point mutation that ablates ATPase activity (E653A). This mutant complex has undetectable ATPase and sliding activities (*Figure 4*) but still binds with a cooperativity and affinity that are indistinguishable from wild type complex (*Figure 4—figure supplement 1*). We then tested the effect of adding ATPase 'dead' complex to wild type at an equal ratio. Binding of this mixture of wild type and mutant complexes is again indistinguishable in terms of either stoichiometry or affinity, confirming that interactions between complexes and with nucleosomes are unaffected by the mutation (*Figure 4—figure supplement 1*). When equal amounts of each complex are present, the distribution of wild type and mutant complexes will reflect that of a Punnett square with wild type:mixed:mutant dimers at ratios of 1:2:1, respectively (*Figure 4C*). The sliding activity of this mixture at 2:1 hINO80: nucleosome showed an initial rate of sliding that was 50% of that of an equivalent amount of wild type complex alone, and the reaction took twice as long to reach completion (*Figure 4D*). Since mutant dimer complexes (25% of total) are inactive in sliding, these account for a 25% reduction in activity. Equally, the wild type dimer complexes (25% of total) can only account for half of the observed activity (i.e. 25% of full wild type rate). Consequently, the remaining half must originate from the heterodimer complexes suggesting that either these dimers have 50% activity of wild type dimers, or that half of them have wild type activity and the other half are inactive. The asymmetry of the DNA lengths flanking the nucleosomes prior to sliding, suggests two possible binding modes for an INO80 dimer such that the active complex is located appropriately to effect sliding in 50% of cases but is not when in the alternative orientation (*Figure 4C*). This proposal would support the latter interpretation of 50% complexes bound productively but will need to be confirmed experimentally. In either scenario, the data show that a single ATPase is both necessary and sufficient to slide nucleosomes. Furthermore, two complexes are still required suggesting that both contribute to sliding even though ATPase activity is required in only one of them.

## Role of the C-terminal domain of hINO80 in coupling ATPase to sliding

The human Ino80 subunit has been shown to contain a C-terminal domain (CTD) that negatively regulates ATPase activity of the hINO80 complex (*Chen et al., 2011*). In order to understand more about the basis of this ATPase regulation and to investigate a role for this in coupling ATPase to sliding, we prepared a complex in which the Ino80 subunit was truncated at residue 1261 to remove the C-terminal domain (hINO80ΔCTD) (*Figure 1—figure supplement 1*). The affinity of this complex for nucleosomes was similar to that of full length complex and two complexes bound to each nucleosome but the binding has lost cooperativity (*Figure 5A*). Two hINO80ΔCTD complexes are still required to effect sliding but the rate has decreased 5-fold compared to hINO80 (*Figure 5B*). By contrast, the nucleosome-stimulated ATPase activity increases two-fold (*Figure 5C*). These two changes result in a ten-fold reduction in coupling efficiency. The cooperativity of sliding and ATPase

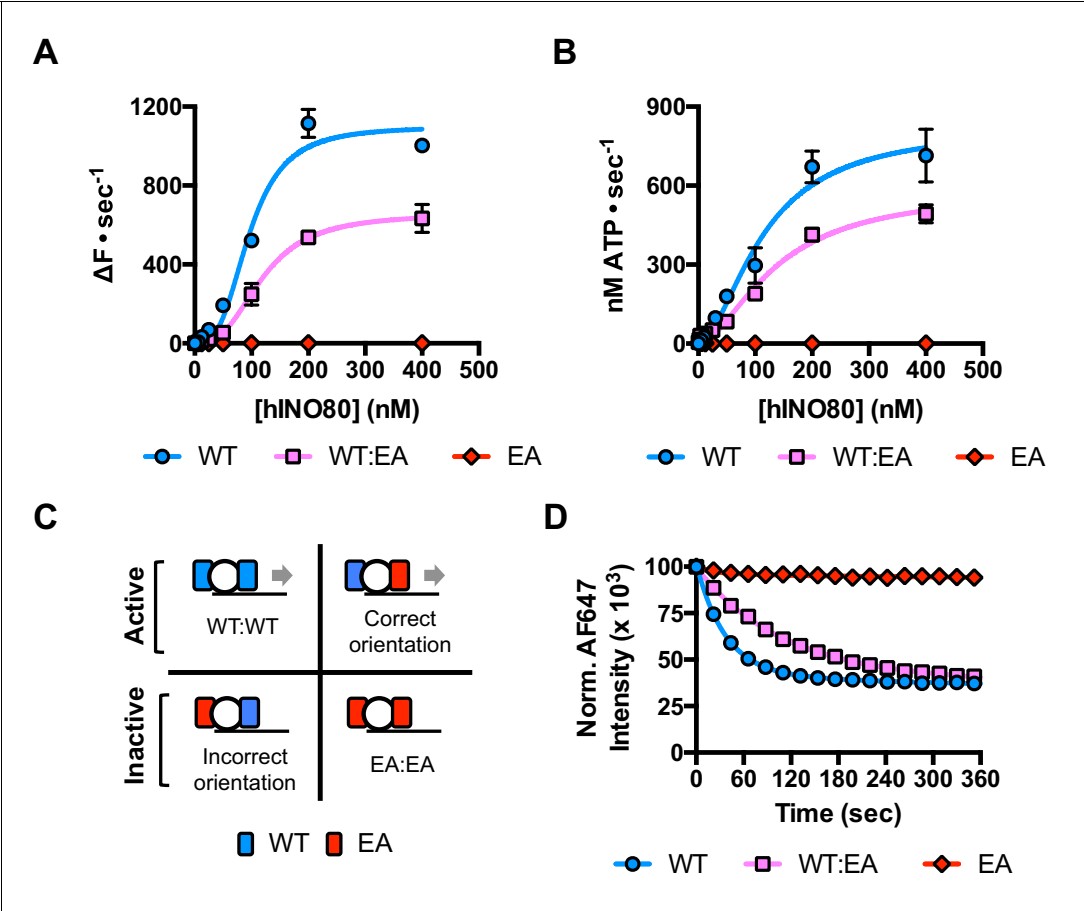

**Figure 4.** A single ATPase is both necessary and sufficient for sliding. (**A**) Concentration dependence of sliding activity for wild type hINO80 (WT), ATPase dead mutant (EA), and an equimolar mixture of the two complexes (WT:EA) at a nucleosome (0N100) concentration of 100 nM. Binding and stoichiometry experiments are shown in *Figure 4—figure supplement 1*. (**B**) ATPase activity under the same conditions as (**A**). (**C**) Punnett square showing distribution of dimer species in a 1:1 ratio mixture. (**D**) Time course of sliding reactions at 2:1 hINO80:nucleosome.

The following figure supplement is available for figure 4:

**Figure supplement 1.** Characterisation of the E653A hINO80 mutant.

remain similar to wild type (*Figure 5—figure supplement 1*). The intrinsic ATPase in the absence of nucleosomes is the same for both hINO80 and hINO80ΔCTD (*Figure 5D*).

An additional consequence of the truncation in the hINO80ΔCTD complex is an effect on the flanking DNA 'sensing' range which is reduced by at least 20 bp (*Figure 6A* and *Figure 6—figure supplement 1*). This reduction of DNA sensing is also reflected in a reduced ability of the complex to slide nucleosomes to a central position (*Figure 6B* and *Figure 6—figure supplement 1*). Finally, unlike hINO80, hINO80ΔCTD slides a centrally positioned 50N50 nucleosome away from the central position (*Figure 6C*). Consequently, we conclude that the hIno80 CTD not only regulates ATPase but also confers both coupling and DNA sensing abilities to the complex.

In order to investigate this role further, we prepared the hIno80 CTD alone (residues 1250–1556) to analyse its intrinsic properties (*Figure 7* and *Figure 7—figure supplement 1*). We find that the isolated CTD is a DNA-binding protein (*Figure 7A*) and is a monomer is solution (*Figure 7B*). Affinity measurements made by MST show that the binding of hIno80CTD to duplex DNA is cooperative with a Hill coefficient close to 2 (*Figure 7C*) but stoichiometry measurements were not possible due to aggregation of the complex at high protein concentrations. However, since two hINO80 complexes are required for sliding and binding, it seems likely that CTD dimerisation is induced by

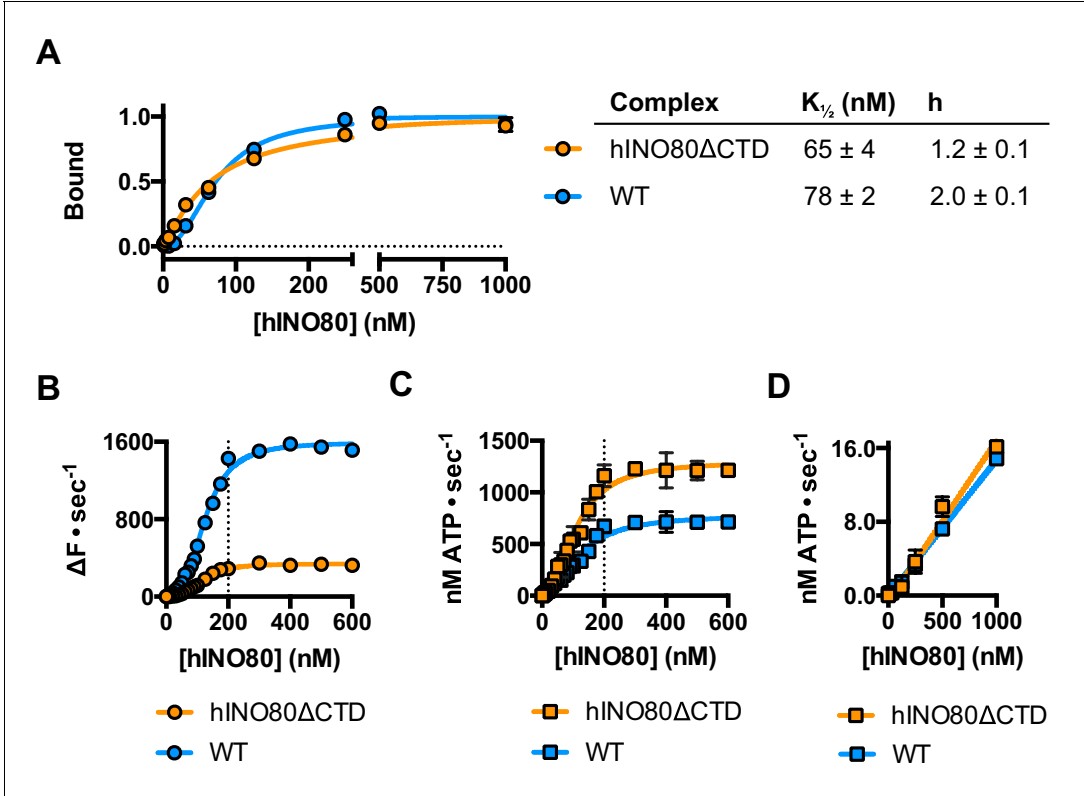

**Figure 5.** hINO80ΔCTD is deficient in ATPase coupling. (**A**) Comparison of nucleosome affinity of hINO80 and hINO80ΔCTD. The nucleosome was 0N100 at a concentration of 20 nM. Hill coefficients and $K_{1/2}$ values are tabulated on the right. Raw MST data are presented in *Figure 5—figure supplement 1*. (**B**) Comparison of protein concentration dependence for sliding activity for hINO80 and hINO80ΔCTD complexes. The maximum rate of sliding for hINO80ΔCTD is approximately 5-fold lower than for hINO80 but is achieved at a 2:1 ratio of complex:nucleosome in both cases. Analysis of the data is presented in *Figure 5—figure supplement 1*. (**C**) Comparison of protein concentration dependence for nucleosome-stimulated ATPase activity for hINO80 and hINO80ΔCTD complex. The maximum ATPase rate for hINO80ΔCTD is approximately 2-fold higher than for hINO80 but is still attained at a 2:1 ratio of complex:nucleosome (0N100) in both cases. Analysis of the data is presented in *Figure 5—figure supplement 1*. (**D**) Dependence of ATPase activity on protein concentration for hINO80 and hINO80ΔCTD in the absence of nucleosomes.

The following figure supplement is available for figure 5:

**Figure supplement 1.** Characterisation of hINO80ΔCTD.

binding to duplex DNA and, like the nucleosome-dependent interaction between hINO80 complexes, this dimerisation is highly cooperative. The CTD, therefore, provides one interface between INO80 dimers although there may be others involving different subunits that we have yet to identify.

Next, we tested whether the hIno80CTD could complement hINO80ΔCTD complexes in *trans* by titrating hIno80CTD into sliding reactions (*Figure 7D*). We found that the sliding activity is indeed stimulated by addition of hIno80CTD, with half maximal stimulation at approximately 1 µM. At the same time, the ATPase rate falls to that of the wild type complex, thereby increasing the coupling of ATPase to sliding, although the effect suggests a weaker binding of the hIno80CTD for this activity inferring that there is more than one binding site for the domain with differing effects. These experiments confirm the coupling role for the hIno80CTD. However, the loss of end sensing resulting from the loss of the CTD cannot be restored in trans (*Figure 7—figure supplement 1*).

Finally, we tested the effect of having a single CTD available in the complex in the context of the ATPase dead motor subunit (E653A) described above. Although the data are complicated due to the mixture of complexes (ratios can be assessed by the Punnett square as discussed above), we simplified the analysis by titrating 1:1 mixtures of a variety of active:inactive complexes with or without a CTD, and assessed them for initial sliding rates and ATPase activities (*Figure 8* and *Figure 8—*

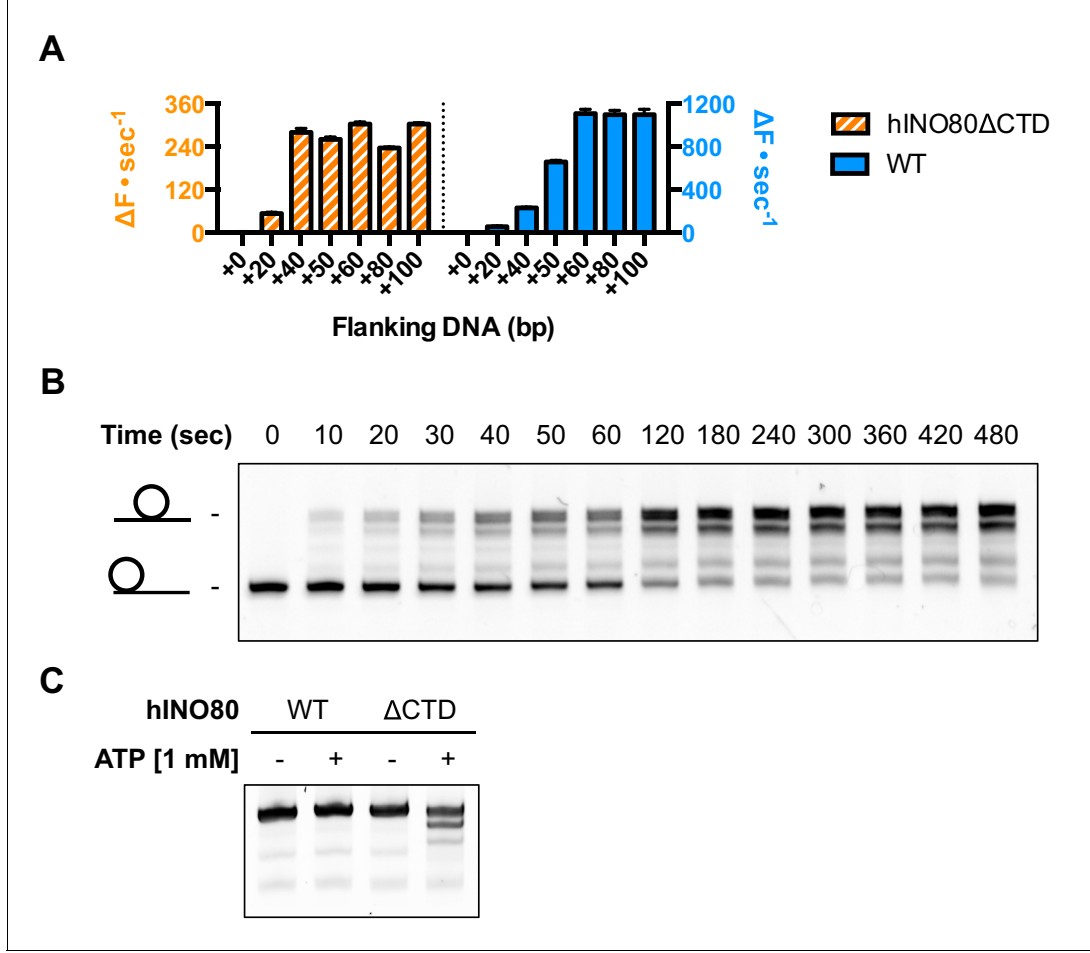

**Figure 6.** hINO80ΔCTD has altered end sensing. (**A**) Initial sliding rates for hINO80 and hINO80ΔCTD on nucleosomes with different length overhangs. Full length hINO80 complex (right) senses DNA ends up to 60 bp while that of the hINO80ΔCTD complex (left) is reduced to under 40 bp. FRET-based assay data are shown in *Figure 6—figure supplement 1*. (**B**) hINO80ΔCTD has a reduced ability to centre nucleosomes. Although the majority of nucleosomes still reach the centre, the spread is much greater than for hINO80 (*Figure 1* and *Figure 3—figure supplement 2*). (**C**) hINO80ΔCTD slides 50N50 nucleosomes away from the central location. These are unaltered by hINO80.

The following figure supplement is available for figure 6:

**Figure supplement 1.** Flanking DNA length dependence of hINO80ΔCTD.

*source data 1*). Combining hINO80 (WT) with the ATPase dead mutant (EA) was shown in *Figure 4* but the data are shown again in *Figure 8* as a reference. The WT:EAΔCTD mixture should have two active dimer components (WT:WT and WT:EAΔCTD in the correct orientation) that each constitute 25% of the complexes. The loss of the CTD in the inactive subunit has a modest overall effect on both ATPase and sliding compared to WT:EA (*Figure 8*). The combined effect of a rise in ATPase with reduced sliding results in a decrease in coupling, along the lines observed for deleting the CTD in the hINO80ΔCTD dimers (*Figure 5*). Cooperativity in sliding was retained (albeit slightly reduced in the WT:EAΔCTD context) but, as for WT:EA, the ATPase activity is non-cooperative (*Figure 8— source data 1*).

By contrast, the combination of WTΔCTD with EA gave more dramatic results (*Figure 8*). In this case, the overall initial sliding rates were similar to WTΔCTD dimers but the cooperativity is now lost. The ATPase activity is similar to WTΔCTD alone and again loses cooperativity, although also continues to increase at super-stoichiometric concentrations of protein so is predicted to plateau at a higher rate. We would expect to see approximately 50% maximal activity compared to WTΔCTD

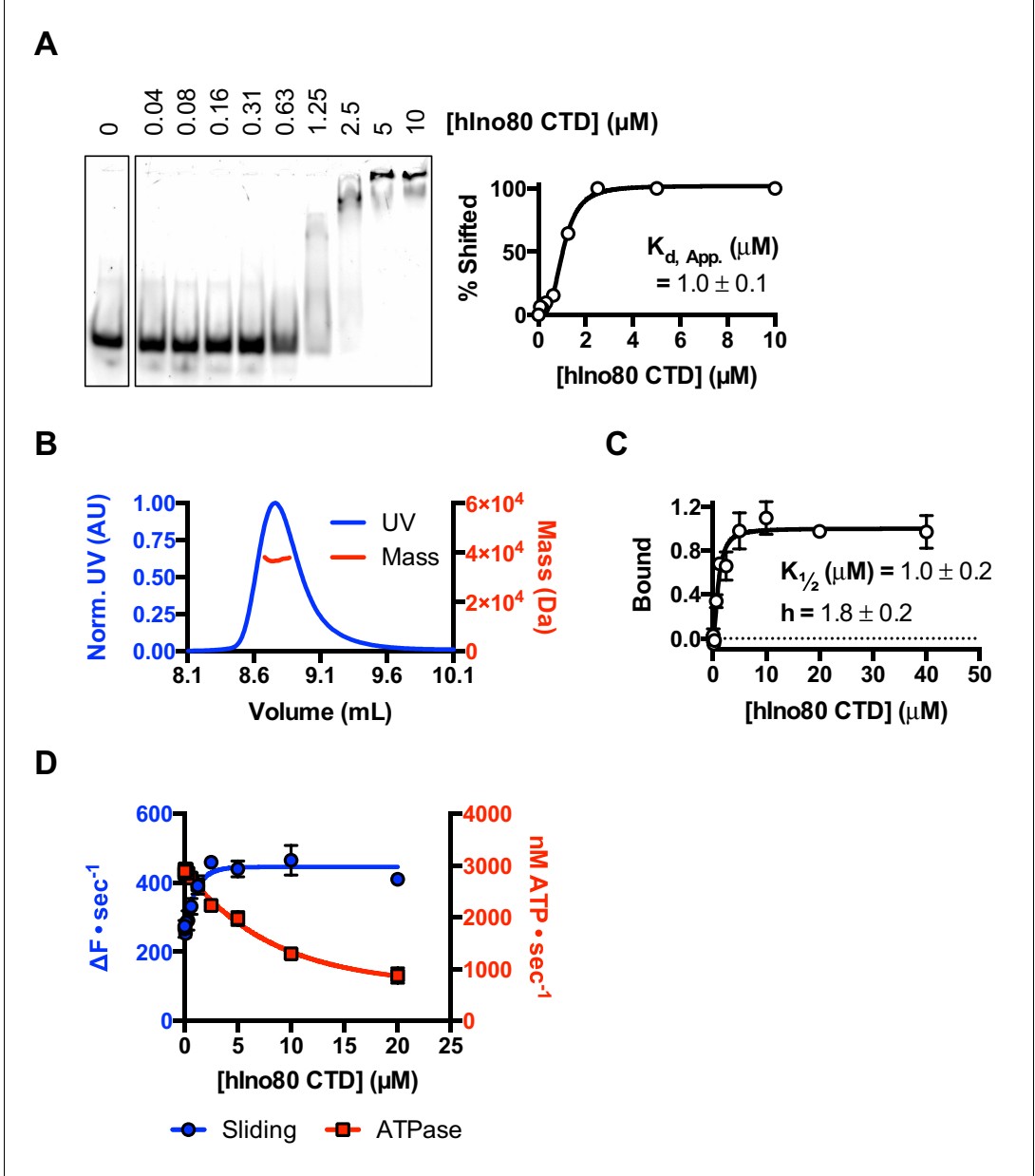

**Figure 7.** The CTD of hIno80 is a cooperative DNA binding protein. (**A**) Gel mobility shift assay using 50 nM 50 bp duplex DNA with increasing concentrations of hIno80CTD. The gel (left) was digitised and band intensity plotted (right) to determine Kd apparent, which is approximately 1 μM. (**B**) SEC-MALS analysis of the hIno80CTD protein (SDS gel shown in *Figure 7—figure supplement 1*). The molecular weight was determined to be 38 ± 1.5 KDa compared to a calculated weight of a monomer of 35.3 KDa. (**C**) Binding of the hIno80CTD to a 50 bp DNA duplex determined by MST. $K_{1/2}$ is 1.0 ± 0.2 μM with a Hill coefficient of 1.8 ± 0.2. (**D**) Effect of titrating in hIno80CTD into a reaction of hINO80ΔCTD (200 nM) with 100 nM 0N100 nucleosomes (compare to *Figure 5*). The ATPase rate decreases but initial sliding rate increases, demonstrating increased coupling of these activities. A $K_d$ apparent for hIno80CTD calculated from the half maximal stimulation is approximately 1 μM. The binding event that regulates ATPase, however, has a lower $K_{1/2}$ that cannot be determined accurately from these data but appears to be at least an order of magnitude weaker.

The following figure supplement is available for figure 7:

**Figure supplement 1.** Characterisation of hIno80CTD protein.

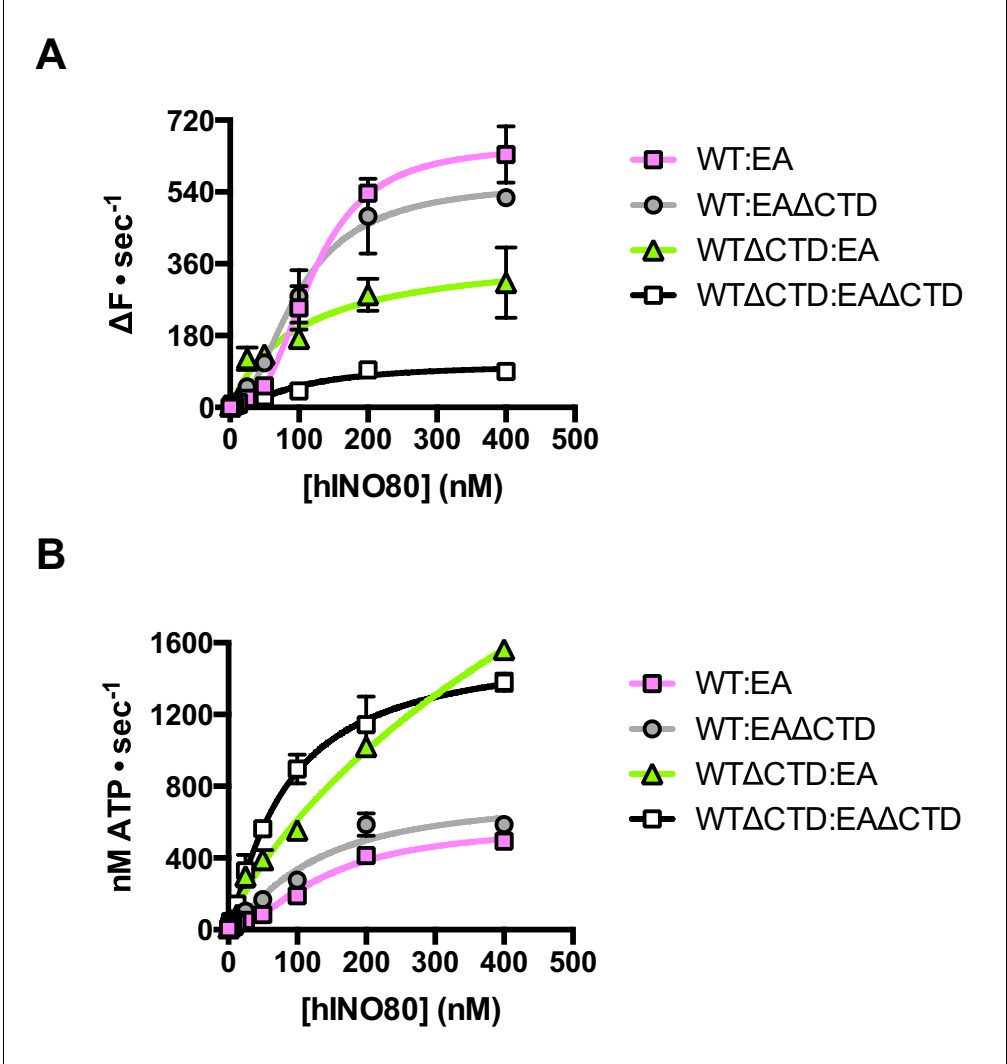

**Figure 8.** The contribution of the hIno80CTD in cis or trans. (**A**) Concentration dependence of sliding activity for equimolar mixtures of different hINO80 complexes as indicated. Complexes are hINO80 (WT), hINO80ΔCTD (WTΔCTD), ATPase dead mutant (EA), and ATPase dead mutant lacking the CTD (EAΔCTD). Assays contained a nucleosome (0N100) concentration of 100 nM. The data are tabulated in *Figure 8—source data 1*. (**B**) ATPase activity under the same conditions as (**A**).

The following source data is available for figure 8:

**Source data 1.** Tabulated maximum rates and Hill coefficients for sliding and ATPase activities of mixtures of INO80 complexes as indicated.

homodimers alone (such as seen for WT:EA mixtures) so this suggests the WTΔCTD:EA hetero-dimers are actually more active in sliding and ATPase than WTΔCTD homodimers.

For completeness, we then examined a mixture of WTΔCTD:EAΔCTD (*Figure 8*). This mixture shows a significant overall fall in sliding activity compared to the WTΔCTD:EA mixture. As before, 50% of the dimers will be WTΔCTD homodimers, so the drop in activity suggests that the hetero-dimers are severely impaired in sliding. ATPase activity of the mixture, by contrast, is similar to WTΔCTD and WTΔCTD:EA at the same protein concentrations so, in the heterodimer, the level of uncoupling is even greater than in WTΔCTD dimers or WTΔCTD:EA heterodimers.

Taken together, these data suggest separate and distinct roles for the CTD contributed by each complex in the dimer that may correlate with the two binding sites on the dimer suggested from our

complementation experiments (*Figure 7*), with one CTD having a principal role in coupling (that seems to need to be connected to the lead motor) while a second, which seems to act in *trans*, regulates ATPase activity of the dimer. A further tier of regulation by the CTD is the end sensing function that requires physical connection to the motor subunit(s). This complex interplay of functions will be best unravelled by structural studies.

## Discussion

Proteins that assist nucleosome sliding are characterised by movement either towards or away from DNA ends. Of those that slide nucleosomes away from ends (or breaks), many show the ability to position nucleosomes at the centre of relatively short pieces of DNA (*Shen et al., 2000*; *Stockdale et al., 2006*; *Udugama et al., 2011*) or to space multiple nucleosomes evenly in arrays (*Längst et al., 1999*; *Udugama et al., 2011*). Nucleosome spacing has been studied extensively in the ISWI family remodellers. ATPase activity of the motor domains of these complexes is regulated by conserved AutoN and NegC sequence motifs and their interactions with H4 histone tails (*Clapier and Cairns, 2012*). These interactions are also involved in the sensing of DNA flanking nucleosomes and centring of them on DNA fragments (*Hwang et al., 2014*). Further data show that the process also involves two ACF complexes (*Racki et al., 2009*) and different nucleotide-dependent conformational states that are also related to binding of H4 tails (*Leonard and Narlikar, 2015*). However, molecular details of how these systems all cooperate in the spacing of nucleosomes are lacking. The only structural information that addresses this is for ISW1a (*Yamada et al., 2011*). These studies suggest that there is a further level of interaction between adjacent nucleosomes that regulates spacing. Consequently, how all of these come together to explain how these remodellers space nucleosomes on DNA remains far from clear.

For the INO80 complex, the understanding is even worse. Although INO80 complex has been shown to space nucleosomes in an apparently similar fashion, the mechanism cannot be the same as ISWI because INO80 lacks AutoN and NegC motifs and ATPase and sliding are not affected by H4 tails (*Udugama et al., 2011*) so cannot be involved in DNA sensing. We now show that, for hINO80, this property is based on an ability to monitor the length of DNA flanking the nucleosome and that this extends to around 50 bp either side, but if the flanking DNA is less than 50 bp then a compromise is reached to centre the nucleosome on the DNA. Instead of regulating the ATPase activity via the H4 tails/AutoN/NegC network, which is lacking in INO80, we show that ATPase is uncoupled rather than inhibited when sliding stops. We also reveal a role for synergistic crosstalk between the C-terminal region of the Ino80 subunit from two complexes in this process both in coupling ATPase to sliding and in DNA sensing. Consequently, although there are apparent similarities at first sight, the molecular mechanism of nucleosome spacing by ISWI and INO80 remodellers is entirely different. As for ISWI, future studies will be required to obtain molecular details of this process.

Obviously, the centring of nucleosomes on short DNA fragments is a wholly artificial system that has little direct relationship to the cellular context but all the same. This uncovers an intrinsic ability of the protein complex that likely has more relevance to the property of nucleosome spacing. However, the involvement of INO80 in the repair of double-stranded DNA breaks (*Shen et al., 2000*; *van Attikum et al., 2004*; *Gospodinov et al., 2011*; *Horigome et al., 2014*) might suggest a role for this function in sliding nucleosomes away from breaks to allow access of the repair machinery. Similarly, the ability to space nucleosomes could relate to the restoration of chromatin structure after repair has taken place.

Cooperativity has been observed for the ACF chromatin remodelling complex (*Racki et al., 2009*), although others like Chd1 are reported to work as monomers (*McKnight et al., 2011*; *Patel et al., 2013*). The functional state of the large, multi-subunit INO80 was unknown, as is that of other large remodellers like RSC (*Cairns et al., 1996*) but structural studies (*Smith et al., 2003*; *Chaban et al., 2008*; Tosi et al., 2013) all show a nucleosome bound to a single complex, which might be expected if samples were prepared with an excess of nucleosomes over remodeller rather than the reverse. We now show that nucleosomes bind two INO80 complexes and sliding by hINO80 involves cooperativity between these two complexes that regulate one another in *trans*. This cooperativity manifests itself in two ways: in binding the nucleosome substrate, and in ATP-dependent nucleosome sliding. Previous studies have suggested a role for the CTD of human INO80 in regulating ATPase activity (*Chen et al., 2011*), although the mechanism of this regulation was

unclear. We now show that the truncated hINO80ΔCTD complex loses cooperativity for nucleosome binding yet still requires binding of two complexes that cooperate to effect sliding, showing that these aspects can be uncoupled and, presumably, are regulated by different contacts between complexes. Furthermore, the isolated CTD is a monomer in solution but binds cooperatively to duplex DNA, most likely forming a dimer interface. The CTD is, therefore, a nucleosome-dependent interface between hINO80 complexes that has a role in regulating binding to nucleosomes and coupling ATPase to sliding, as well as in sensing the distance from DNA ends. However, the interface responsible for cooperative sliding remains undetermined. Given the importance of the CTD in human INO80, it is perhaps surprising that, although conserved in higher organisms, the CTD does not appear to be conserved in yeast INO80 (*Chen et al., 2011*). However, studies of the yeast Arp8 protein have revealed a dimerisation interface that is not present in human Arp8 (*Saravanan et al., 2012*),could provide an alternative means for association of yeast INO80 complexes.

We and others have begun to reveal the roles of accessory subunits in INO80, finding functions for coupling (Arp5 and Ies6) and regulation of ATPase activity (Ies2) amongst the conserved core of subunits (*Chen et al., 2013*; *Watanabe et al., 2015*; *Willhoft et al., 2016*; *Yao et al., 2016*). It has also been shown that the Arp4 and Arp8 subunits are involved in histone recognition (*Downs et al., 2004*; *Fenn et al., 2011*; *Gerhold et al., 2012*; *Saravanan et al., 2012*). For the ISWI family remodellers, nucleosome sliding is regulated by sequence motifs that compete with a binding site for the H4 tails (*Clapier et al., 2001*; *Hamiche et al., 2001*; *Dang et al., 2006*; *Clapier and Cairns, 2012*), although it is not clear whether this is in cis within a complex or in trans between complexes that interact. By contrast, INO80 shows very little dependence on binding of nucleosome tails and is able to slide tailless nucleosomes as well as cognate ones (*Udugama et al., 2011*). Instead, our data show that regulation is through interaction between complexes in *trans*, and this is a requirement to position nucleosomes away from DNA ends. The cooperativity is the result of nucleosome-mediated physical contact between complexes via their C-terminal domains, although we cannot rule out additional communication between two monomers bridged by contacts through the nucleosome. Indeed, the finding that nucleosome sliding by the hINO80ΔCTD complex retains cooperativity, despite losing cooperativity for binding, suggests strongly that there are other contacts involved between complexes.

However, why two complexes of INO80 are required for nucleosome sliding is not intuitively apparent given that the machinery to carry out nucleosome sliding resides within a single subunit, such as in Chd1, where a monomer is also thought to be sufficient for sliding (*McKnight et al., 2011*; *Patel et al., 2013*) even though two complexes bind to each nucleosome (*Nodelman et al., 2017*). However, other remodellers, such as ACF, have a requirement for a dimer (*Racki et al., 2009*) suggesting this is some form of regulation rather than an intrinsic requirement for activity. It has been suggested that asymmetry in the nucleotide-dependent conformational states in the dimer are linked to DNA length sensing (*Leonard and Narlikar, 2015*) although other studies show a link between the H4 tail binding and the AutoN and NegC with DNA sensing (*Hwang et al., 2014*). Our data suggest that nucleosome sliding by hINO80 only requires one active (or coupled) ATPase motor, but that motor subunit is regulated by its partner in the dimer. A similar situation has been reported for ACF (*Leonard and Narlikar, 2015*). A Chd1 monomer senses flanking DNA across the gyres of the nucleosome thereby sensing the flanking side closest to the location of the bound motor (*Nodelman et al., 2017*). Two wild type hINO80 complexes acting cooperatively could sense both ends simultaneously in a similar manner. However, to then slide the nucleosome, either one complex slides in reverse to work in synergy with the other, or one of them becomes inactivated or uncoupled. Indeed, given that the dimer senses DNA flanking length on both sides of the nucleosome simultaneously, presumably one complex performs the sensing role on one side while the other monitors the other side. The presence of a short DNA flank (or an end) would be signalled to the active motor and uncouple ATPase from sliding. The roles for each complex could readily be swapped if the direction of sliding were reversed. One sensing protomer must therefore regulate its partner in trans to attain directionality, through control of coupling of ATPase to sliding rather than by regulating the ATPase activity itself.

This model implies that a complex deficient in its primary ATPase activity (stemming from the motor domains) could stimulate the activity/coupling of an active INO80 if its sole role were to sense DNA ends (e.g. resulting from a double-strand break) and then activate its partner to move away from this end. Our data show that a single ATPase activity is indeed sufficient to permit sliding of

nucleosomes from an end to the centre of DNA fragments. Furthermore, when using mixed ATPase dead and wild type complexes, we observe no signs of 'overshoot' (where the sliding has gone beyond the midpoint and then returned), suggesting a slow stop as the nucleosome reaches the end point of the reaction resulting from continuous monitoring of flanking DNA length either side of the nucleosome even when one of the motors is inactive. This would be consistent with the observation that initial sliding rates are slower for nucleosomes with shorter flanking sequences, as also seen for ACF (*Yang et al., 2006*). Furthermore, continuous monitoring of ATPase during, and beyond, a sliding reaction by hINO80 shows the rate does not alter over this time period revealing that the slow stop must be due to a gradual uncoupling of ATPase from sliding. However, since we observe that ATPase activity merely becomes uncoupled from sliding as the nucleosome reaches the middle, then it would make no difference whether the partner was inactivated or uncoupled from sliding, as both scenarios would produce the same result.

The single subunit remodeller Chd1 has a modular structure with several domains of defined functions including a C-terminal DNA binding domain (*Delmas et al., 1993*). Recent negative stain electron microscopy data for Chd1 (*Nodelman et al., 2017*) shows that the DNA-binding domain is located on the flanking DNA, albeit on no more than one side of the nucleosome at a time. Similarly, ISWI contains a C-terminal DNA-binding domain with the same fold as that of Chd1 (*Grüne et al., 2003*; *Sharma et al., 2011*) which likely plays a similar role. Our data now reveal that a previously unidentified DNA-binding domain is located in a similar position in the human Ino80 subunit as well, although we find no detectable sequence homology or similarity in secondary structure prediction with the equivalent domains of either Chd1 or ISWI, nor with any other known DNA-binding domain. Interestingly, the ACF remodeller shows altered sensing of flanking DNA associated with the Acf1 subunit (*Yang et al., 2006*) although the location of this subunit on bound nucleosomes is currently unknown. These functional similarities suggest a common DNA sensing mechanism between these otherwise distinct remodeller families. Structural studies on a related enzyme ISW1a raise another possibility - that a single protein complex might interact with a dinucleosome (*Yamada et al., 2011*).

An intriguing observation from our experiments is that ATPase rate is unaltered whether the hINO80 is actively sliding nucleosomes or not. Although INO80 contains multiple subunits in addition to the motor domains of the Ino80 subunit that are capable of binding and/or hydrolysing ATP (actin, Arp4, Arp5, Arp8, Tip49a, Tip49b), it is revealing to note that the INO80E653A mutant has undetectable ATPase activity even when bound to nucleosomes (*Figure 4B*) showing that all of the ATPase activity we measure derives from the motor domains. Although the notion that ATPase activity of INO80 bound to nucleosomes is constant and continuous may be counter-intuitive, the rate is actually very low (1–2 s$^{-1}$) compared to other SF2 helicases (*Singleton et al., 2007*). Consequently, the amount of ATP consumed is small in comparison with the overall energy balance within a cell and, whatever the function of this may be, it appears to be worth paying the metabolic price. However, it is important to note that ATPase activity is only stimulated when INO80 is bound to nucleosomes so when chromatin becomes compacted, access is restricted and INO80 (presumably) is released. It is therefore possible that, when chromatin becomes unpacked for DNA repair after detection of a double-stranded DNA break, a constant monitoring of nucleosome location, by a bound and active INO80 complex near the repair site, would provide continuous access for repair machinery until the repair has been effected, after which the chromatin structure could be restored and INO80 released.

## Materials and methods

### Preparation of recombinant hINO80 complexes

Human Ino80 genes lacking the coding sequence for the N-terminal sub-domain (residues 1–266, NTD) or the NTD and C-terminal sub-domain (residues 1262–1556) were generated by PCR and then sub-cloned to append TEV protease-cleavable N-terminal 8-histidine and C-terminal twin Strep-tag II tags. The tagged subunit was then cloned with other genes of the hINO80 core complex using a modified MultiBac system (*Berger et al., 2004*) as described previously (*Willhoft et al., 2016*). Ino80 subunit DEAQ-box mutant (E563A) was prepared by InFusion-based mutagenesis. Catalytically inactive hINO80 core complexes and wild type sub-complex were expressed and purified in the same manner as the wild type complex (*Willhoft et al., 2016*).

## Preparation of nucleosomes

Nucleosomes with varying flanking DNA lengths were prepared using a similar method described in (*Willhoft et al., 2016*). For centred nucleosome substrates, histone octamers were formed into nucleosome core particles (NCPs) using a 184 bp DNA fragment based on the Widom601 sequence, with pre-digested AvaI and HinfI sticky ends. Sticky end compatible oligonucleotides were then ligated onto this to generate nucleosomes with equivalent DNA lengths flanking the central NCP.

## Gel-based sliding assay

Gel-based sliding assays were essentially carried out as described previously (*Willhoft et al., 2016*). Briefly, for end-point assays using nucleosomes with varying lengths of flanking DNA, recombinant hINO80 complex was incubated with nucleosome at a 2:1 molar ratio for 30 min at 37°C before adding ATP and $MgCl_2$ to final concentrations of 1 and 2 mM, respectively. For time course experiments, a master mix of hINO80 and nucleosome was incubated at 37°C for 30 min before adding 18 µL of reaction mix to 2 µL of 10 mM ATP and 20 mM $MgCl_2$ (pre-aliquoted into separate tubes for each time point). Reactions were typically run in reverse time-order, with the longest time point started first, unless time intervals were less than 15 s. For all gel-based assays, reactions were stopped by adding a molar excess of plasmid DNA or unlabelled nucleosomes, as well as EDTA (pH 8.0) to a final concentration of 2 mM, and incubating at room temperature for 15 min before resolving the reaction products on a 6% acrylamide gel prepared and run in 0.5x TBE.

## Binding studies of hINO80 and nucleosome by microscale thermophoresis (MST)

Measurements for equilibrium binding were performed using 20 nM fluorescently-labelled nucleosomes as previously described (*Willhoft et al., 2016*) with minor changes: experiments were conducted at 25°C and in 'MST Buffer' [25 mM HEPES pH 8.0, 50 mM NaCl, 1 mM TCEP, 10% glycerol, 0.01% Tween-20 and 0.1 mg/mL BSA], with LED and MST power set at 80%. Binding curves were fitted to two sets of replicates.

For binding experiments, data were analysed in the NT Analysis software (NanoTemper) using a 'Hill method' analysis on temperature jump data. Hill coefficients were confirmed by generating a Hill plot and calculating a best-fit line. Data points at the extremes of the range were excluded from the analysis.

For saturation point determination, fluorescent nucleosomes (labelled on histone H3 on R3 with AlexaFluor 647 (Thermo Fisher Scientific)) were kept at a constant concentration of 500 nM. Runs were conducted with an LED power of 5% and MST power set to 40%. Straight lines were fitted to the unsaturated and saturated portions of the data in triplicate using GraphPad Prism 6.0f. The stoichiometry was extrapolated from the intersection of the two lines using the following equation:

$$[Ino80 : Nucleosome] = \frac{\left(\frac{Y_1 - Y_2}{S_2 - S_1}\right)}{[N]}$$

Where $Y_1$ and $S_1$ are the Y-intercept and slope of the unsaturated data fit, $Y_2$ and $S_2$ the same parameters for the saturated fit, and [N] the nucleosome concentration.

## Analysis of hINO80:nucleosome complexes by electron microscopy

hINO80 and 18N18 nucleosomes were mixed to final concentrations of 150 and 75 nM, respectively, in buffer containing [25 mM HEPES, 60 mM KCl, 0.5 mM TCEP] and $ADP-BeF_3$ [10 mM NaF, 2 $BeCl_2$, 2 mM $MgCl_2$, 2 mM ADP], and incubated at 37°C for 30 min. 2 µl of this reaction mix was deposited for 1 min on glow-discharged grids with continuous carbon. The excess liquid was blotted away and 2 µl of 2% uranyl acetate solution was added for 1 min. The stain was blotted away and the grids left to dry.

Images were collected on a CM200 Philips microscope operating at 200 kV and equipped with a TVIPS slow scan $4 \times 4k$ CCD camera. Micrographs were manually collected in low dose mode (~20 electrons / $\text{Å}^2$ / sec) at 38000x nominal magnification resulting in a pixel size of 2.14 Å. A few hundred particles were picked manually using EMAN (*Ludtke et al., 1999*) and subjected to 2D

classification with IMAGIC (*van Heel et al., 1996*). The best classes were used as templates for automated particle picking with Gautomatch software (K. Zhang, unpublished).

The data were binned to a final pixel size of 4.28 Å. A first round of classification was performed with RELION (*Scheres, 2012*) in order to sort the particles. This resulted in ~6800 good particles. These were submitted to alignment and 2D classification, which ended with ~3000 particles distributed in' large complex' classes (corresponding to dimers bound to a nucleosome) with the remainder in 'small complex' classes (mainly free complexes or nucleosomes). In order to improve the class details, the best particles were submitted to multi-reference alignment and classification with SPIDER software (*Frank et al., 1996*) and IMAGIC (20 to 40 particles per class).

## Measurement of initial velocities of nucleosome sliding and ATPase by hINO80

Sliding and ATPase activities were measured as described previously (*Willhoft et al., 2016*), and performed in a one-pot reaction in the presence of 100 nM dual fluorescently-labelled nucleosome substrate (unless otherwise stated in the text). For unstimulated ATPase activity, hINO80 WT or ΔCTD complexes were titrated as indicated. The raw traces for sliding activity were scaled to 100,000 FU before further analysis. For ATPase rates, ADP standard curves for an NADH concentration of 200 μM were measured simultaneously with each experiment to determine the absolute amount of ATP used per unit time.

For titration of hINO80 WT or ΔCTD complexes with 100 nM nucleosome substrate, initial velocities ($V_{0, Sliding}$ and $V_{0, ATPase}$) at each concentration of hINO80 complex were determined manually in GraphPad Prism and then fitted using the 'Allosteric sigmoidal' model under default settings, which in turn calculated a theoretical $V_{max}$. To obtain cooperativity coefficients, the data were linearised based on this $V_{max}$ and plotted as a function of log([hINO80]); straight line fits were then applied to the data between asymptotes by excluding data at the extremes of the range. The slope and error associated with this fit were then extracted as Hill coefficients.

To measure the ATPase activity of hINO80 upon reaching an apparent equilibrium in the sliding reaction, 400 μM NADH was used instead of 200 μM in the coupled-assay reaction mix. This allowed us to monitor the signal change for a prolonged period. Data points were re-scaled as a percentage for illustrative purposes.

For experiments comparing WT and catalytically inert hINO80 complexes (carrying a point mutation of E653 to alanine in the conserved D<u>E</u>AQ-box), WT, mutant or a 1:1 molar mix of WT and mutant hINO80 complexes were prepared to a final total protein concentration of 2 μM. This stock was then serially diluted to produce a 5x working stock for each reaction condition in the titration. The remaining assay conditions were as described (*Willhoft et al., 2016*). All reaction mixes were allowed to equilibrate at 37°C for at least 30 min before manually dispensing reaction mixtures into 384-well micro-titre plates in duplicate.

## Preparation of recombinant Ino80 CTD (1250–1556)

Domain boundaries for the human Ino80 CTD were determined using sequence alignments and secondary structure prediction. The gene construct was obtained by PCR from a codon optimised human Ino80 gene carrying the coding sequence for C-terminal twin Strep-tag II tags, and then subcloned into pMAL using InFusion to additionally append the coding sequences for an N-terminal MBP tag. The resulting plasmid was transformed into B834(DE3) cells already carrying a rare codon plasmid and plated onto LB agar with 100 μg/mL ampicillin and 34 μg/mL chloramphenicol. A single colony was picked into 100 mL of L broth with the appropriate antibiotics; this was used to inoculate larger volumes of LB media for expression purposes. Large cultures of transformant cells were grown to an $OD_{600}$ of 0.6 at 37°C and expression induced by the addition of IPTG to a final concentration of 1 mM. Expression was allowed to continue for at least 16 hr at 18°C.

Purification for Ino80 CTD followed a four step purification protocol. Cells were harvested by centrifugation at and resuspended in 'CTD Buffer A' [25 mM HEPES, 300 mM NaCl, 10% glycerol, pH 8.0], with the addition of 1 mM EDTA and one tablet of cOmplete, EDTA-free Protease Inhibitor Cocktail (Roche) per litre of culture. Cells were disrupted in a high pressure homogeniser (AVESTIN) and the lysate clarified by centrifugation at 40,000 x g for 1 hr. The clarified lysate was then passed through a 0.45 μm filter before being loaded onto a 5 mL StrepTactin HP column (GE Healthcare),

which was pre-equilibrated in 'CTD Buffer A'. After loading and washing, the sample was eluted with 5 mM desthiobitin into 1 mL fractions.

Peak fractions were pooled and the sample loaded directly onto a 1 mL Heparin HP column (GE Healthcare) pre-equilibrated in 'CTD Buffer A'. All bound material was eluted into small fractions by a 10 column volume gradient from 300 mM-1 M NaCl, with Ino80 CTD eluting at approximately 600 mM NaCl. After identifying the appropriate fractions to combine by SDS-PAGE, the fractions were pooled and the protein concentration estimated using the extinction coefficient of the MBP-Ino80 CTD-2xSII construct at 280 nm. In addition to 1 mM DTT, a 1:20 ratio of His$_6$-tagged tobacco etch virus (TEV) protease was then added to remove all affinity tags; this reaction was allowed to proceed at room temperature for 1–2 hr.

The uncleaved material, cleaved MBP affinity tag and TEV protease were separated from Ino80 CTD by passing the protease reaction mix over a 5 mL MBPTrap HP (GE Healthcare) and 1 mL cOmplete His-Tag Purification Column (Roche), both of which were pre-equibrated in 'CTD Buffer A'. From here the sample was concentrated and applied to a Superdex 75 (GE Healthcare) column also equilibrated in 'CTD Buffer A'. Peak fractions were pooled, concentrated and frozen in liquid nitrogen in small aliquots, before being stored at −80°C until further use.

## Preparation of 50 bp dsDNA

Complementary top and bottom strand oligos to generate a 50 bp duplex were mixed in a 1:1 molar ratio in 'annealing buffer' [25 mM Tris pH 7.5, 50 mM NaCl, 0.5 mM EDTA]. Annealing reactions were carried out in 200 µL volumes by first heating to 95°C and gradually cooling to 30°C over 1 hr. The annealed duplex was then quantified by absorbance at 260 nm. To generate labelled duplexes, the top strand was replaced with a 5′-FAM (6-carboxyfluorescein)-labelled oligo of identical sequence. All duplex stocks were maintained at concentrations greater than 100 µM.

## Binding studies with Ino80 CTD and dsDNA by EMSA and MST

Binding of Ino80 CTD and 50 bp duplex DNA was analysed by electrophoretic mobility shift assay (EMSA) and MST. For analysis by EMSA, Ino80 CTD was serially diluted at 2x final concentration in 'EMSA Buffer' [25 mM HEPES, 50 mM NaCl, 10% Glycerol, 0.5 mM EDTA, pH 8.0] and then mixed with an equal volume of 2x concentration 50 bp dsDNA, prepared as described above. The final reaction contained 50 nM 50 bp DNA with varying concentrations of Ino80 CTD as indicated. The reaction mixes were incubated at room temperature for 1 hr before resolving the reactants by native gel electrophoresis (6% acrylamide, 0.5x TBE).

MST experiments were carried out at 25°C in the same buffer used for experiment with hINO80 complex and nucleosomes. Briefly, Ino80 CTD was serially diluted at 2x final concentration and then mixed with an equal volume of FAM-labelled 50 bp duplex DNA in 'MST Buffer'. These mixes were then equilibrated at room temperature for 1 hr before loading into Monolith NT.115 MST Premium Coated capillaries (NanoTemper). Thermophoresis binding data were analysed in GraphPad Prism using the 'Specific binding with Hill slope' model under default settings.

## SEC-MALS with Ino80 CTD

Purified Ino80 CTD was used for molecular weight analysis by size exclusion chromatography-multiangle light scattering (SEC-MALS) with a Superdex 75 (GE Healthcare) size exclusion column. The column was extensively equilibrated in 'MALS Buffer' [25 mM HEPES, 300 mM NaCl] at room temperature before applying a 100 µL of Ino80 CTD through a capillary loop. The system flow was maintained at 0.5 mL/min. Light scattering and UV absorbance data at 280 nm were collected across a 25 mL volume post-injection using a miniDAWN TREOS light scattering detector (Wyatt). All data were exported and re-plotted in GraphPad Prism.

## Titration of Ino80 CTD into hINO80ΔCTD complex

Measurements of nucleosome sliding and ATPase activities in the presence of increasing concentrations of recombinant Ino80 CTD were carried out simultaneously as described previously for the WT and ΔCTD complexes alone. It was found, however, that the Ino80 CTD had to be diluted into the reaction mix last. hINO80ΔCTD and 0N100 nucleosomes were therefore incubated together for 15 min at 37°C together with the coupled-ATPase assay components. Subsequently, a 5x reaction stock

of Ino80 CTD at the required concentration was added to each reaction condition, and the sample mixed by slow pipetting. The final reaction mix contained 200 nM hINO80ΔCTD, 100 nM dual-fluorophore labelled 0N100 nucleosomes, 1 mM phosphoenolpyruvate, 200 U/mL pyruvate kinase, 40 U/Ml lactate dehydrogenase and 200 μM NADH, in a buffer of 25 mM HEPES, 50 mM NaCl and 1 mM TCEP, along with varying concentrations of Ino80 CTD (from 20 μM to a lowest concentration of 20 nM).

## Acknowledgements

We thank D Rhodes for providing the plasmid containing the 167 bp Widom sequence and S. Halford for providing highly purified EcoRV. We would also like to thank C Mckeown for providing guidance on using the SEC-MALS apparatus and L Yates for useful discussions on MST. The work was funded by the Wellcome Trust (DBW and XZ) and Cancer Research UK (DBW).

## Additional information

### Funding

| Funder | Grant reference number | Author |
| --- | --- | --- |
| Cancer Research UK | C6913/A21608 | Dale B Wigley |
| Wellcome | 095519/Z/11/Z | Dale B Wigley |
| Wellcome | 098412/Z/12/Z | Xiaodong Zhang |

The funders had no role in study design, data collection and interpretation, or the decision to submit the work for publication.

### Author contributions

OW, Conceptualization, Formal analysis, Investigation, Writing—original draft, Writing—review and editing; EAM, RB-D, LO, Investigation; RJA, Formal analysis, Investigation; XZ, Supervision, Funding acquisition, Project administration; DBW, Conceptualization, Supervision, Funding acquisition, Writing—original draft, Writing—review and editing

### Author ORCIDs

Oliver Willhoft, http://orcid.org/0000-0003-2422-1839
Xiaodong Zhang, http://orcid.org/0000-0001-9786-7038
Dale B Wigley, http://orcid.org/0000-0002-0786-6726

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
