## [Decision Letter]

Thank you for submitting your article "Cross-talk within a functional INO80 complex dimer regulates nucleosome sliding" for consideration by *eLife*. Your article has been reviewed by two peer reviewers, and the evaluation has been overseen by Jerry Workman as the Reviewing Editor and James Manley as the Senior Editor. The reviewers have opted to remain anonymous.

The reviewers have discussed the reviews with one another and the Reviewing Editor has drafted this decision to help you prepare a revised submission.

Summary:

Previously, it was shown that the INO80 remodeler generates evenly spaced nucleosome arrays. Here, Wigley and co-workers investigate the spacing properties of INO80, and show that the remodeler cooperatively binds to both sides of the nucleosome. While cooperative binding is analogous to the ACF remodeler, INO80 has a distinct architecture, and the authors present a convincing case that the regulatory mechanism responsible for DNA sensing is unrelated to other remodelers studied so far.

In addition to dimerization on the nucleosome, a major finding is that the C-terminal domain is responsible for both cooperative nucleosome binding and also efficiently coupling ATP hydrolysis to nucleosome sliding. Remarkably, they show that the C-terminal domain provided in trans also has a stimulatory effect. This emphasizes how the INO80 system is distinct from that of other remodelers.

Essential revisions:

Concerns that need to be addressed prior publication:

1) Do the major conclusions of this work hold for the full (not minimal) Ino80 complex?

2) Subsection “hINO80 dimers slide nucleosomes cooperatively”, last paragraph: The authors seem to favor possibility (iii) which is a major conclusion of the work and stated in the Abstract. They provide evidence against possibility (i), but I'm not convinced that possibility (ii) has been ruled out. Possibility (ii) seems to be the likeliest, and in fact is also what has been seen for ISWI enzymes.

3) Sliding gels: the authors make arguments throughout the paper whether their complexes are able to precisely position the nucleosome at the center. Quite a good resolution is necessary to make these claims, which the sliding gels lack. Minimally, proper mobility standards should be run on the same gel (for example, run an unremodeled 50N50 nucleosome next to the 0N100 sliding reaction to better show that the end product of sliding is 50N50). Running thermally shifted nucleosomes on a separate(!) gel is not sufficient (Figure 1—figure supplement 1). The authors may want to find a better and more accurate way to determine the precise location of the nucleosome along the DNA (e.g. by footprinting, crosslinking etc.). Otherwise the authors must qualify their conclusions regarding the resolution of the remodeling assays.

4) The authors propose that the CTD of one remodeler is required to couple sliding with hydrolysis for the other remodeler bound to the nucleosome. They present an experiment where ATPase dead proteins are mixed with wildtype, finding the rate to be 50% that of normal. These results are supportive of the model the authors describe. While their experiment with addition of isolated CTD demonstrate that the CTD can act in trans (and thus could come from the other remodeler bound to the nucleosome), it might be informative to perform a related experiment where the ATPase-active remodeler lacks the CTD and the ATPase-dead remodeler has the CTD; in this way, CTD can only contribute by helping the active ATPase. Since they show the CTD to dimerize, such an experiment (which would only provide one CTD per heterodimer) could also help add insight into how the two remodelers work together. Although it may be hard to interpret rates for such a mixture, it might still be informative to look at the length of DNA sensed (as in Figure 6).

5) The authors nicely show that the CTD in trans can impact ATPase activity and sliding. I think it would be worthwhile to also report what happens when isolated CTD is added to INO∆CTD in terms of length sensing. Figure 6 shows clearly that the remodeler is unable to sense more than 40 bp without the CTD, and it would helpful to reveal whether the effects reported in Figure 7 are also coupled to DNA length sensing.

---

## [Author Response]

*Essential revisions:*

*Concerns that need to be addressed prior publication:*

*1) Do the major conclusions of this work hold for the full (not minimal) Ino80 complex?*

We presume the “full” complex refers to that which has been identified by pull down experiments from human cells? This complex consists of six additional subunits that are unique to mammalian INO80 and are not conserved in yeast which has a whole set of different additional proteins that are unique to yeast INO80. By contrast, the subunits that comprise our “minimal” core are conserved across INO80 complexes from all species and includes all of the potential ATPase subunits. Where similar studies have been conducted, the properties we show for our human core complex (e.g. nucleosome spacing, sliding activity) are similar to those reported by others for both human and yeast “full” complexes (e.g. Conaway lab (human), Bartholemew, Peterson & Wu labs (yeast)). Indeed, the Conaway lab has shown that optimal sliding activity for human INO80 complex requires a large excess of complex over nucleosomes which would be consistent with a need to saturate with dimers rather than monomers, although they only use very small amounts of enzyme and substrates in their assays, presumably due to low availability of complex because the human INO80 complex is present at very low abundance in human cells. To our knowledge, the only lab that has made sufficient quantities of human INO80 full complex for in vitro assays is the Conaway lab. They have published a protocol to prepare FLAG-tagged complex from human cells but the yields are so poor that they require analysis by silver-stained gels or Western blotting and the homogeneity of those preparations is unclear. The Conaway lab also showed that removal of the N-terminal 265 residues of human INO80 subunit resulted in complete loss of the additional non-conserved subunits from protein prepared from human cells. This complex (essentially the same as our core complex) had unaltered nucleosome sliding or ATPase activity from that of the full complex. It was therefore deduced that these components do not contribute to enzyme activity per se but may be involved in regulation. The additional subunits in the full complex have diverse, non-conserved functions across different species so it seems unlikely that these will, for example, allow the complex to function by a completely different mechanism (as a monomer, for example) or have a different regulatory system that no longer involves the C-terminal domain, which are the main conclusions of our work. Furthermore, the stoichiometry of the additional proteins is unknown and nor is it known if all six can bind simultaneously to the complex.

Nonetheless, we have made some “full” complex by expressing full length human INO80 subunit in insect cells along with the usual subunits of the core plus the six additional proteins. The complex we prepare in this way now contains 5 of the additional 6 proteins (MCRS1 either does not express in insect cells or does not associate with the complex) although is not as high quality or yield as our core complex. Nevertheless, this (almost) full complex behaves essentially the same as the core complex (Figure 1—figure supplement 1). The binding of nucleosomes, sliding rates and ATPase activities are similar to the core complex and exhibit similar cooperativity. In fact, the sliding rate for the core complex is higher than for the full complex so, if anything, shows better coupling. Short of repeating every experiment in the manuscript, we hope this now provides convincing evidence that the core and full complexes do indeed behave similarly.

*2) Subsection “hINO80 dimers slide nucleosomes cooperatively”, last paragraph: The authors seem to favor possibility (iii) which is a major conclusion of the work and stated in the Abstract. They provide evidence against possibility (i), but I'm not convinced that possibility (ii) has been ruled out. Possibility (ii) seems to be the likeliest, and in fact is also what has been seen for ISWI enzymes.*

The argument here is to distinguish between uncoupled ATPase and a rapid back and forwards oscillation around the midpoint. We can start by pointing out that the ATPase can easily be uncoupled from sliding by loss of the Arp5 subunit (published by us and several other groups). Furthermore, we show here (as seen for ACF) that the ATPase rates are similar with short overhangs and with longer overhangs even though initial sliding rates are slower so must be uncoupled, at least to some extent. In fact, nucleosomes with no overhangs and which, therefore, cannot slide, still stimulate ATPase which is completely be uncoupled from sliding. Therefore, the complex is certainly able to uncouple ATPase from sliding under a variety of different circumstances. This is in contrast to ISWIa that the Bartholemew lab has shown not to be stimulated by centrally positioned nucleosomes nor is it able to move them (a point now made and referenced in the revised manuscript). The same lab also showed that yeast INO80 cannot slide centrally located nucleosomes but has ATPase activity stimulated by them (already stated in the original manuscript but now with greater emphasis).

However, to obtain further evidence for our proposal that Ino80 stops sliding and becomes uncoupled, we have carried out an additional experiment (Figure 1—figure supplement 3). Using our method to add overhangs to either side of nucleosomes, we have engineered nucleosomes that are offset from the centre by defined numbers of base pairs. We used these as standards on gels to determine the precise location of the final product. Since we are able to resolve these standards which differ by 2bp in their location, we can assess the spread of the products. The tightness of the band suggests a single (or limited number of) product(s) although we cannot rule out some product spaced one or two bases either side, but if this were the case then both the switching and the rate of sliding would have to be very fast, indeed faster than sliding towards that point and the distribution of bands would reflect that. Although we cannot rule that out, it seems unlikely. We have therefore added some comments to address this in the revised manuscript. This experiment is also relevant to the point below.

*3) Sliding gels: the authors make arguments throughout the paper whether their complexes are able to precisely position the nucleosome at the center. Quite a good resolution is necessary to make these claims, which the sliding gels lack. Minimally, proper mobility standards should be run on the same gel (for example, run an unremodeled 50N50 nucleosome next to the 0N100 sliding reaction to better show that the end product of sliding is 50N50). Running thermally shifted nucleosomes on a separate(!) gel is not sufficient (Figure 1—figure supplement 1). The authors may want to find a better and more accurate way to determine the precise location of the nucleosome along the DNA (e.g. by footprinting, crosslinking etc.). Otherwise the authors must qualify their conclusions regarding the resolution of the remodeling assays.*

Actually, we do not argue that INO80 positions nucleosomes exactly at the centre of fragments, although that claim has been made by others. Instead, our proposal is that the complex positions nucleosomes with respect to the length of flanking DNA sequence and this is only centrally located if the flanks are less than 50-60bp on each side, due to simultaneous DNA sensing on either side of the nucleosome. We completely agree that the positioning of the bands cannot be determined with single base pair resolution on gels and nowhere did we imply that the positioning was that accurate. In fact, on the contrary, it seems most probable that there will be some “slip” either side by at least a few base pairs. That being said, our bands are very sharp showing a limited spread of positions and certainly much less than 10bp either side which is distance we can detect readily in our gel system, as demonstrated by the heated nucleosome sample. Our use of this “marker” (incidentally, clearly labelled as being from a separate gel so not a marker) was more to demonstrate the separation distance that would be expected for a 10 base pair difference rather than as an accurate measure for other gels and showing this is clearly discernable in our gels. We do indeed use appropriate centrally located markers on gels in Figure 1 and Figure 6 in the original manuscript and now, most pertinently, also in Figure 1—figure supplement 2 of the revised manuscript. In fact, this definition of “centrally positioned” nucleosomes has been used in other publications (e.g. from the Narlikar & Bartholemew labs and several others) on the basis of positions on gels and has been confirmed directly by crosslinking data for yeast INO80 (Bartholemew lab). However, the thermal sliding shows a series of preferred positions that would be consistent with 10 base pair (i.e. single turn) increments.

Our gels show a tight, single band at the centre of the fragment. There is no evidence of any significant spread and the distribution appears to be within a base pair or two as larger displacements would be evident from their position. This favours either our hypothesis for a stationary, uncoupled nucleosome or that, at worst, the nucleosome is oscillating no more than a few base pairs either side of this position. Since the ATPase rate does not alter once the nucleosome has reached the middle this is only possible if the directional switching is significantly faster than the ATPase rate or this would be detected in the ATPase measurements. While we cannot rule out this unlikely scenario, neither can we confirm it. Consequently, we have now added a comment to this effect in the revision.

*4) The authors propose that the CTD of one remodeler is required to couple sliding with hydrolysis for the other remodeler bound to the nucleosome. They present an experiment where ATPase dead proteins are mixed with wildtype, finding the rate to be 50% that of normal. These results are supportive of the model the authors describe. While their experiment with addition of isolated CTD demonstrate that the CTD can act in trans (and thus could come from the other remodeler bound to the nucleosome), it might be informative to perform a related experiment where the ATPase-active remodeler lacks the CTD and the ATPase-dead remodeler has the CTD; in this way, CTD can only contribute by helping the active ATPase. Since they show the CTD to dimerize, such an experiment (which would only provide one CTD per heterodimer) could also help add insight into how the two remodelers work together. Although it may be hard to interpret rates for such a mixture, it might still be informative to look at the length of DNA sensed (as in Figure 6).*

This experiment is an interesting suggestion which we had considered but we shared the concern that the results might be difficult to interpret. Nonetheless, we have now done this experiment (Figure 8 and [Supplementary-material SD1-data]). We decided that the only way to be able to compare these activities in any interpretable way would be to titrate in a 50:50 mixture of the two complexes (WT△CTD and EA) so that we could estimate the contribution made by each possible dimer pair to the overall mixture. We should have 25% of dimers as “dead” EA homodimers and 25% as active WT△CTD homodimers. The remaining 50% are WT△CTD: EA heterodimers, half of which are bound in a “productive” mode. The sum of these activities would suggest, as for similar experiments already shown for WT:EA mixtures, that the overall rate should be 50% of that of WT△CTD alone. However, the results were not so simple. First, the sliding activity actually peaks at a very similar rate to that of just WT△CTD alone. Second, this activity is no longer cooperative although still peaks at two complexes per nucleosome. Thirdly, the ATPase activity is now completely non-cooperative and appears to require a super-stoichiometric level for maximal activity. As anticipated, these data are hard to interpret but one interesting feature is that the data suggest the addition of a single CTD from the EA complex in trans to the WT△CTD subunit actually enhances the sliding activity. Unfortunately, the complexities of this mixture preclude us being able to obtain interpretable data regarding end sensing.

However, in light of these new data we carried out two additional experiments not requested by the reviewers (Figure 8 and [Supplementary-material SD1-data]). The first of these was to test the effect of deleting both CTDs and also adding an ATPase dead subunit into the mix (WT△CTD:EA△CTD). This mixture was very poor in sliding but retained high ATPase activity so has become even more uncoupled.

The second experiment was to titrate in an equal mixture of WT core complex (with CTD) and the EA△CTD. This again provides a single CTD in the heterodimer but in the reverse context to that above. These experiments showed that presence of a single CTD in this context had little effect on either the sliding rate or its cooperativity. The ATPase rate was marginally enhanced showing reduced coupling. These results differ quite dramatically from those with the WT△CTD:EA described above. This suggests that, rather than CTDs contributing to sliding in trans between complexes, the lead complex requires its own CTD. The second CTD from a partner seems to have some role in regulating the ATPase of the active sliding partner but behaves differently when there is no partner CTD in the active complex.

We have added an additional section in the Results describing these experiments which have provided intriguing new insight into the role of the CTD and we thank the reviewers for pointing us in this direction.

*5) The authors nicely show that the CTD in trans can impact ATPase activity and sliding. I think it would be worthwhile to also report what happens when isolated CTD is added to INO∆CTD in terms of length sensing. Figure 6 shows clearly that the remodeler is unable to sense more than 40 bp without the CTD, and it would helpful to reveal whether the effects reported in Figure 7 are also coupled to DNA length sensing.*

We agree regarding the complementation of INO80△CTD by CTD in regard to length sensing and we now include in this modified version of the manuscript (additional supplementary figure panel (Figure 7—figure supplement 1)). This experiment shows that even though the CTD, when added in trans, is able to re-confer some coupling of ATPase to sliding to the Ino80△CTD complex, it is not able to restore the end sensing. Presumably there is some, as yet unidentified, contribution that requires a physical linking of the CTD to the complex.